# Re-visiting the *trans* insertion model for complexin clamping

**Shyam S Krishnakumar[1][*][†], Feng Li[1][†], Jeff Coleman[1], Curtis M Schauder[1], Daniel Kümmel[1,2], Frederic Pincet[1,3], James E Rothman[1], Karin M Reinisch[1][*]**

[1]Department of Cell Biology, Yale University School of Medicine, New Haven, United States; [2]School of Biology/Chemistry, Univeristät Osnabrück, Osnabrück, Germany; [3]Laboratoire de Physique Statistique, UMR CNRS 8550 Associée aux Unive, Ecole Normale Supérieure, Paris, France

**Abstract** We have previously proposed that complexin cross-links multiple pre-fusion SNARE complexes via a *trans* interaction to function as a clamp on SNARE-mediated neurotransmitter release. A recent NMR study was unable to detect the *trans* clamping interaction of complexin and therefore questioned the previous interpretation of the fluorescence resonance energy transfer and isothermal titration calorimetry data on which the *trans* clamping model was originally based. Here we present new biochemical data that underscore the validity of our previous interpretation and the continued relevancy of the *trans* insertion model for complexin clamping.

## Introduction

The tightly regulated release of neurotransmitters is key to all information processing in the neural circuitry. The fusion of a synaptic vesicle to release the neurotransmitters is mediated by the SNARE (Soluble N-ethylmaleimide-sensitive factor Attachment protein REceptor) complex, which forms between vesicle and target membranes as v-SNAREs emanating from transport vesicles assemble with t-SNAREs emanating from target membranes (*Sollner et al., 1993*; *Weber et al., 1998*; *Jahn and Scheller, 2006*). Key proteins regulating SNARE-mediated fusion at the synapse are the calcium sensor synaptotagmin and complexin (CPX) (*Brose et al., 1992*; *McMahon et al., 1995*; *Fernandez-Chacon et al., 2001*; *Giraudo et al., 2006*; *Sudhof and Rothman, 2009*). Genetic and physiological studies in a number of model systems show that CPX inhibits the spontaneous release of neurotransmitters and is also essential for synchronous exocytosis (*Huntwork and Littleton, 2007*; *Maximov et al., 2009*; *Yang et al., 2010*; *Martin et al., 2011*; *Cho et al., 2014*). CPX 'clamps' the SNARE assembly process to prevent the continuous release of neurotransmitters (*Giraudo et al., 2006*). It does so by stabilizing the SNAREs in an otherwise unavailable 'intermediate' energetic state in which the four helix bundle is about 50% zippered (*Li et al., 2011*).

Based on the X-ray crystal structure of CPX bound to a mimetic of this half-zippered intermediate in which only the N-terminal portion (residues 26–60) of v-SNARE, VAMP2, is present (SNAREΔ60), we proposed a molecular model for the clamping of the SNARE assembly by CPX (*Kümmel et al., 2011*). We found that the CPX central helix (CPX_cen, the SNARE-binding domain) binds one SNAREpin while the accessory helix (CPX_acc, the clamping domain) extends away and bridges to a second SNAREpin. The CPX_acc interacts with the t-SNARE in the second SNAREpin, occupying the v-SNARE binding site, thus inhibiting the full assembly of the SNARE complex. Further, the intermolecular *trans* clamping interaction of CPX organizes the SNAREpins into a 'zig-zag' topology that is incompatible with opening a fusion pore (*Krishnakumar et al., 2011*; *Kümmel et al., 2011*).

We used isothermal titration calorimetry (ITC) to characterize the interaction of the CPX_acc with the t-SNARE, fluorescence resonance energy transfer (FRET) analysis to establish the angled conformation

*For correspondence: shyam.krishnakumar@yale.edu (SSK); karin.reinisch@yale.edu (KMR)

†These authors contributed equally to this work

Competing interests: The authors declare that no competing interests exist.

**eLife digest** Molecules called neurotransmitters are used to carry signals between neurons. The neurotransmitters in the first neuron are stored in small bubble-like structures called synaptic vesicles. When this neuron is ready to send a signal to a second neuron, the membrane that encloses the synaptic vesicle fuses with the cell membrane that surrounds the neuron. This involves SNARE proteins in the vesicle membrane interacting with similar proteins in the cell membrane to form a SNARE complex, which then proceeds to 'zip' the two membranes together.

Other proteins are involved in the fusion process and the release of the neurotransmitters. For example, complexins bind to SNARE proteins during the formation of the SNARE complex in order to temporarily halt the fusion process. This 'clamping' interaction ensures that the neurotransmitters are released at the appropriate time.

Researchers have proposed two different models of the clamping interaction. In the *trans* clamping model a region in the complexins called the accessory helix extends forward and clamps SNARE proteins that are present on the two membranes. An alternative model explains clamping in terms of electrostatic interactions between the accessory helix and the two membranes. These interactions are repulsive because the accessory helix and the membranes are all negatively charged.

Now Krishnakumar, Li et al.—including some of the researchers who first proposed the *trans* clamping model—have used a variety of biochemical techniques to re-examine the clamping interaction. These experiments support the idea that the accessory helix binds to and clamps a SNARE protein, as suggested by the *trans* clamping model. The results of recent in vivo experiments on fruit flies have also provided support for the *trans* clamping model, although further work is need to compare the models in both in vitro and in vivo systems.

of CPX$_{acc}$ which allows the *trans* clamping interaction, and the cell–cell fusion assay (*Hu et al., 2003*) to functionally test the zig-zag model for CPX clamping (*Krishnakumar et al., 2011*; *Kümmel et al., 2011*). Recently, Rizo, Rosenmund, and colleagues (*Trimbuch et al., 2014*) have re-examined the clamping interaction of CPX and have raised concerns regarding the interpretation of the ITC and FRET data and the use of the cell–cell fusion assay as an in vitro system to study CPX clamping (*Krishnakumar et al., 2011*; *Kümmel et al., 2011*). Here we address these concerns and argue that the *trans* clamping model we had previously proposed remains relevant.

## Results

### ITC experiments

In our earlier paper we used ITC experiments to confirm that the CPX$_{acc}$ interacts with the t-SNARE in the truncated pre-fusion SNARE complex (SNAREΔ60) as predicted by the X-ray crystal structure (*Kümmel et al., 2011*). To measure this interaction, we blocked the central helix binding site by pre-binding the SNAREΔ60 complex with a truncated form of CPX (CPX-48; residues 48–134) before titration. In the recent report by *Trimbuch et al. (2014)* the authors suggest that the 1.5 molar excess of the CPX-48 that was used to block the CPX$_{cen}$ binding does not saturate the central helix binding site and the heat observed upon addition of CPX to blocked SNAREΔ60 arises from the completion of CPX$_{cen}$ binding rather than from interactions involving the CPX$_{acc}$. This was primarily based on their ITC data which showed that CPX-47 (CPX47–134) binds to truncated complex SNAREΔ60 with an affinity constant ($K_d$) = 2.39 ± 0.19 μM and to non-truncated SNARE complex with $K_d$ = 339 ± 9 nM (*Trimbuch et al., 2014*). The binding constant for full-length CPX and a non-truncated SNARE complex is reported to be ∼20 nM (*Pabst et al., 2002*) and, given that CPX-48 has an intact central helix including all SNARE-interacting residues (residues 48, 52, 69, and 70) (*Chen et al., 2002*), the expectation would be that CPX-48 and full-length CPX bind to the non-truncated SNARE complex with similar affinities. This discrepancy prompted us to repeat their ITC experiments and, under our experimental conditions, the $K_d$ for CPX-48 binding to the post-fusion SNARE complex was 43 ± 7 nM (*Figure 1A*, *Table 1*), much closer to the value reported for the full-length CPX (*Pabst et al., 2002*). We found further that CPX-48 bound the pre-fusion SNAREΔ60 with a $K_d$ = 457 ± 47 nM (*Figure 1B*, *Table 1*), or about five times more tightly than reported by *Trimbuch et al. (2014)*. These binding

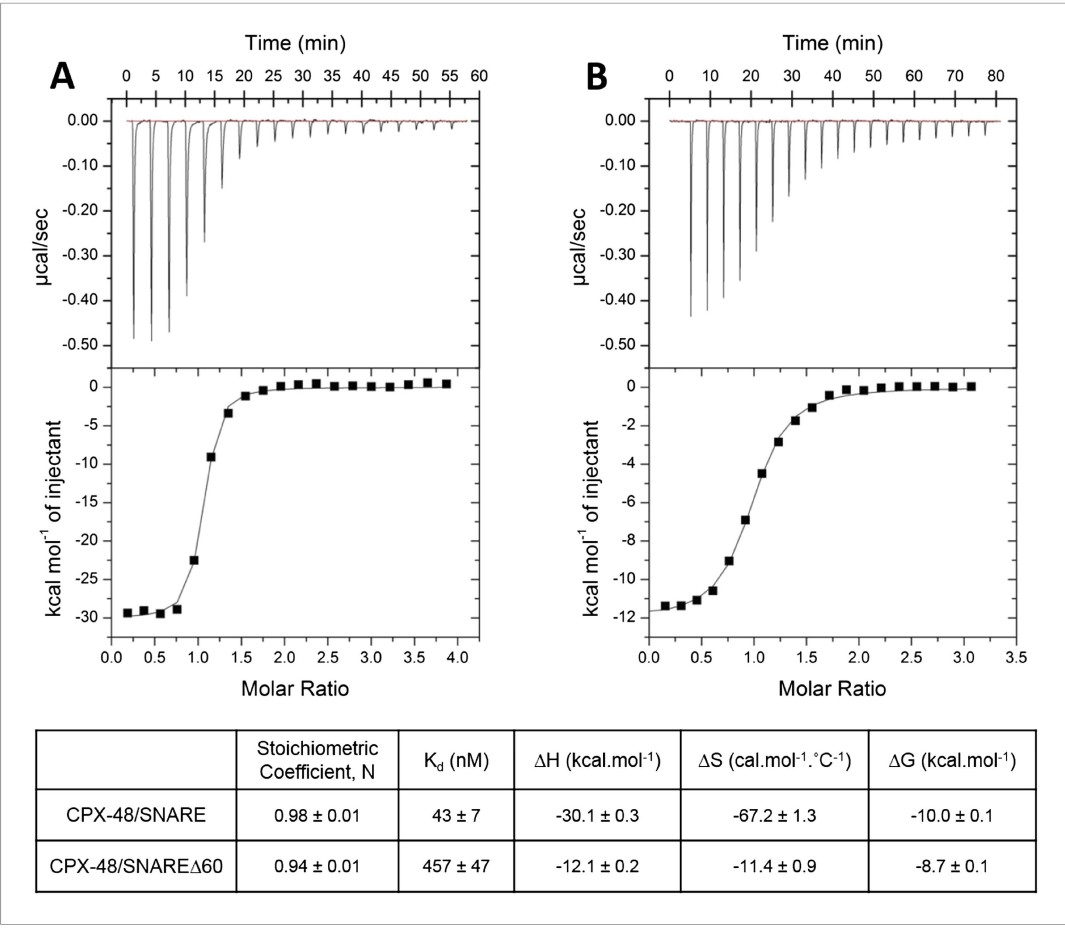

| | Stoichiometric Coefficient, N | $K_d$ (nM) | $\Delta H$ (kcal.mol$^{-1}$) | $\Delta S$ (cal.mol$^{-1}$.°C$^{-1}$) | $\Delta G$ (kcal.mol$^{-1}$) |
|---|---|---|---|---|---|
| CPX-48/SNARE | 0.98 ± 0.01 | 43 ± 7 | -30.1 ± 0.3 | -67.2 ± 1.3 | -10.0 ± 0.1 |
| CPX-48/SNAREΔ60 | 0.94 ± 0.01 | 457 ± 47 | -12.1 ± 0.2 | -11.4 ± 0.9 | -8.7 ± 0.1 |

**Figure 1**. Characterization of interaction of complexin (CPX)-48 with pre- and post-fusion SNARE complex by isothermal titration calorimetry. Representative thermograms of CPX-48 titrated into post-fusion SNARE (**A**) or pre-fusion SNAREΔ60 complex (**B**). The solid lines represent the best fit to the corresponding data points using non-linear least squares fit with the one-set-of-sites model. The results of the fit from 2–3 independent trials are shown in *Table 1*.

constants ensure the near saturation (~96%) of the central helix binding site under the conditions (1.5 molar excess of CPX-48) reported in *Kümmel et al. (2011)*. Thus, with the blocked SNAREΔ60, only the interaction between CPX$_{acc}$ and t-SNARE would be measured. This is supported by the fact that the interaction affinity could be modulated by the mutations in CPX$_{acc}$. Hydrophobic mutations (D27L, E34F, R37A) increased the binding affinity (approximately eightfold stronger than the wild-type, *Table 1*), while the introduction of charged residues (A30E, A31E, L41E, A44E) abolished the interaction (*Kümmel et al., 2011*).

To restore confidence in our ITC data reported in *Kümmel et al. (2011)*, we also repeated the ITC binding experiments using 2.5–3-fold molar excess of CPX-48 to completely block the CPX$_{cen}$ binding (≥99%). We found that CPX binds to the blocked SNAREΔ60 with a binding affinity of 15.2 ± 1.4 μM (*Figure 2A*, *Table 1*), matching well the K$_d$ ~16 μM reported in *Kümmel et al. (2011)*. We note that in these experiments as well as those reported in Kümmel et al. we titrated full-length CPX (residues 1–134) into the blocked SNAREΔ60, and not the minimal functional domain (residues 26–83) as we had implied ("we used a complexin construct comprising both the central and accessory helices [residues 26–83]" [*Kümmel et al., 2011*]), and we apologize for this reporting error. We have now additionally carried out the ITC experiments with the minimal functional domain (CPX26–83) and find that this truncated version also binds to the blocked SNAREΔ60, albeit with slightly weaker affinity (K$_d$ = 23.9 ± 0.1 μM) compared with full-length CPX (*Figure 2B*, *Table 1*). Taken together, the data strongly support our earlier conclusion that the ITC binding studies carried out with CPX titrated into

**Table 1**. Affinity constants ($K_d$) for complexin (CPX) binding to SNARE complexes measured by isothermal titration calorimetry

| Complexin | Binding partner | Binding affinity ($K_d$) | Reference |
|---|---|---|---|
| CPX1–134 | Ternary SNARE | 19 nM | *Pabst et al. (2002)* |
| CPX48–134 | Ternary SNARE | 43 ± 7 nM | This study |
| CPX48–134 | SNAREΔ60 | 457 ± 47 nM | This study |
| CPX1–134 | Blocked SNAREΔ60 (1.5-fold excess of CPX48–134) | 16 µM | *Kümmel et al. (2011)* |
| Super-clamp CPX1–134 (D27L E34F R37A) | Blocked SNAREΔ60 (1.5-fold excess of CPX48–134) | 2 µM | *Kümmel et al. (2011)* |
| CPX1–134 | Blocked SNAREΔ60 (3-fold excess of CPX48–134) | 15.2 ± 1.4 µM | Current study |
| CPX26–83 | Blocked SNAREΔ60 (3-fold excess of CPX48–134) | 23.9 ± 0.1 µM | Current study |

blocked SNAREΔ60 correctly reflect the binding of $CPX_{acc}$ to t-SNARE. Consistent with this, we have recently also been able to characterize the binding of mammalian $CPX_{acc}$ to *Drosophila* t-SNAREs using blocked *Drosophila* pre-fusion SNARE complex (*Cho et al., 2014*). Mutations in the $CPX_{acc}$ predicted to enhance or decrease the binding of $CPX_{acc}$ to t-SNARE exhibit corresponding binding profiles in ITC experiments (*Cho et al., 2014*) in support of the *trans* insertion model (*Kümmel et al., 2011*).

Further, to directly monitor the multiple binding modes of CPX to the pre-fusion SNAREpin, we carried out new ITC experiments where we titrated full-length CPX into unblocked SNAREΔ60. As our initial ITC data had suggested that the accessory helix of super clamp CPX (residues 1–134 with D27L, E34F, R37A; scCPX) binds to SNAREΔ60 with ~10× higher affinity (*Table 1*) than wild-type (*Kümmel et al., 2011*), we used scCPX for this analysis. As shown in *Figure 3A*, titration of scCPX into the unblocked SNAREΔ60 results in a thermal graph characteristic of a reaction involving multiple binding sites, demonstrating that CPX has more than one binding site per SNAREΔ60 complex. The data can be best approximated using the independent thermodynamic parameters for the $CPX_{cen}$ and $CPX_{acc}$ interaction, with the assumption that both the truncated SNAREΔ60 and scCPX are bivalent (*Figure 3B*). We observed a qualitatively similar titration curve for wild-type CPX (*Figure 3C*) but, since the $CPX_{acc}$ interaction with SNAREΔ60 is much weaker, the fitting with multiple binding sites was not resolved in detail.

We suspect that experimental factors such as the buffer conditions, purification methodology (i.e., presence or absence of affinity tags), and the method used to determine the protein concentration (Bradford/BCA or $A_{280}$) might contribute to the variability in the ITC data between the two papers. The latter is of particular significance since the quantitation of small proteins/peptide (<7 kDa) is highly dependent on the method used. However, we are unable to pinpoint the differences in protocol between the two studies since several relevant experimental details (e.g., whether affinity tags were present or what method was used for protein quantification) are not described in Trimbuch et al. Another discrepancy with regard to the ITC measure may derive from the way the blocked SNAREΔ60 complex was assembled. In our experience, assembly of SNARE complex with VAMP60 or similarly truncated VAMP using only the concentration–dilution cycles as carried out in the Trimbuch et al. report results in heterogeneous samples, with non-productive aggregates and un-assembled components not very effectively removed by the concentration cycle. In our experimental regimen we always purified the truncated SNARE complex on a Superdex 75 gel filtration column to ensure good quality of the assembled complexes. Column-purified SNAREΔ60 complexes were subsequently incubated with gel-filtration purified CPX-48 to form the blocked SNAREΔ60 complex used in the ITC experiments. We also note that *Pabst et al. (2002)* used a more stringent purification protocol (Mono-Q purification) even though they used a different SNAP25 construct for the preparation of the SNARE complexes for their ITC studies and report $K_d$ values similar to our findings (*Pabst et al., 2002*). The samples used by Trimbuch et al. for the NMR analysis appear to be homogeneous, however.

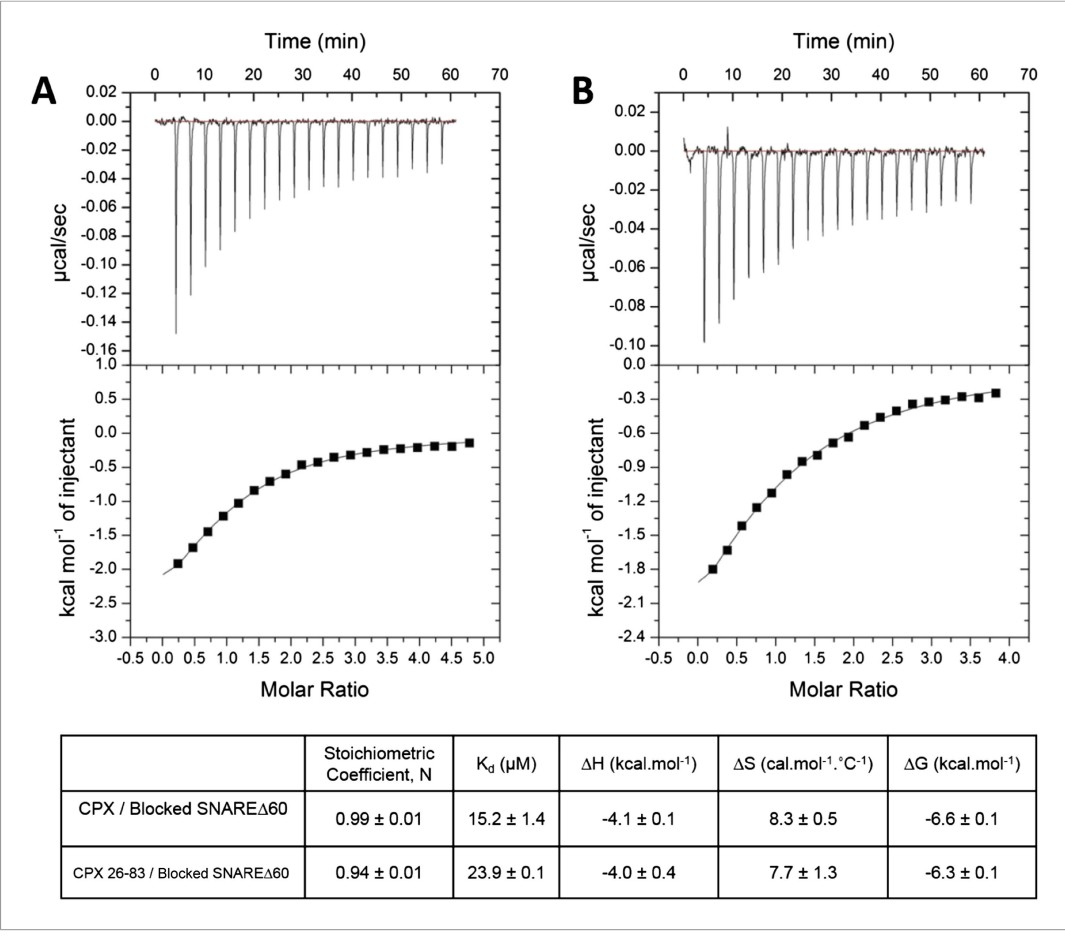

**Figure 2**. Interaction of complexin accessory helix (CPX$_{acc}$) with the t-SNARE groove for full-length and truncated (residue 26–83) CPX characterized by isothermal titration calorimetry. Full-length (**A**) or CPX26–83 (**B**) were titrated into pre-fusion SNAREΔ60 complex with the CPX central helix (CPX$_{cen}$) binding site blocked with CPX-48 to exclusively measure the CPX$_{acc}$–t-SNARE clamping interaction. The solid lines represent the best fit to the corresponding data points using non-linear least squares fit with one-set-of-sites-model and results of the fit are shown in *Table 1*. All experiments were conducted in triplicate and a representative thermogram is shown.

## FRET analysis

We used FRET analysis to establish that the angled conformation of CPX$_{acc}$ also occurs in solution and is not dictated by crystal packing (*Kümmel et al., 2011*). We placed the donor probe on SNAP25, acceptor probe on CPX$_{acc}$, and used donor quenching to track the positioning of the CPX$_{acc}$ in the pre- and post-fusion SNARE complex (*Krishnakumar et al., 2011*; *Kümmel et al., 2011*). The FRET data were consistent with the CPX$_{acc}$ locating parallel to the SNARE complex in the fully-assembled SNARE complex, but moving away from the SNAREs in the pre-fusion half-zippered complex (*Krishnakumar et al., 2011*; *Kümmel et al., 2011*). In *Trimbuch et al. (2014)*, based on NMR analysis, it was stated that the CPX$_{acc}$ helix is poorly structured even when bound to the SNARE complex and exhibits higher flexibility with c-terminal truncation of VAMP. The authors therefore suggested that the low FRET state we observed in the pre-fusion SNAREpin and assign to the 'angled' conformation can be explained by the enhanced flexibility of the CPX$_{acc}$ in this complex. Even though this is not in contradiction to the *trans* clamping model we have proposed, there are several lines of evidence (*Krishnakumar et al., 2011*; *Kümmel et al., 2011*) arguing against this interpretation. (1) We observe a well-defined 'high' and 'low' FRET state for CPX bound to the post- and pre-fusion SNARE using two different FRET pairs with different $R_0$ values (Stilbene/Bimane, $R_0$ ~27Å and Bimane/Oregon Green, $R_0$ ~38Å), and the FRET distances in these CPX–SNARE complexes match very well with the predicted

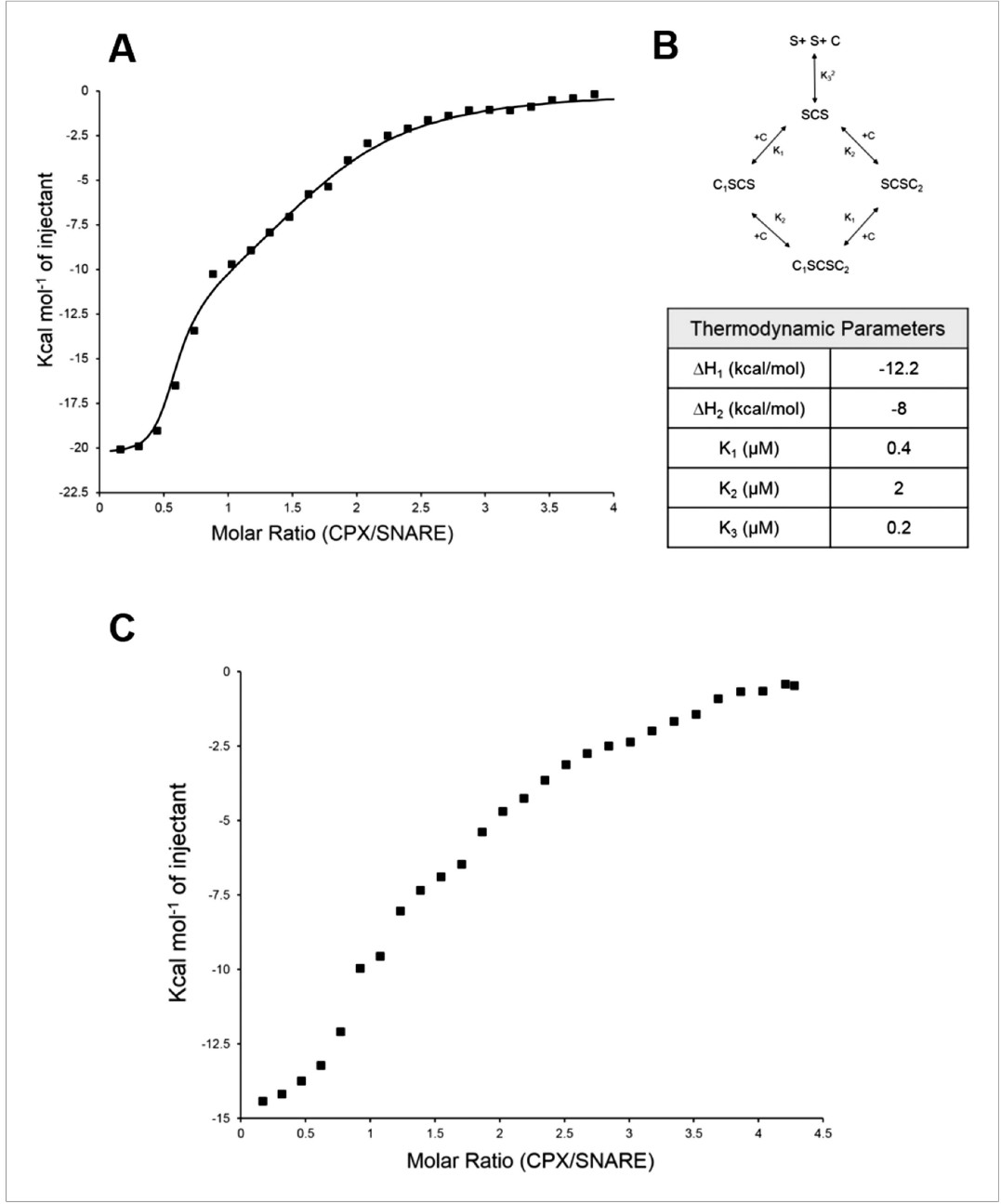

**Figure 3**. Isothermal titration calorimetry indicates multivalent interactions between SNAREΔ60 and CPX. (**A**) Calorimetric titration of super-clamp complexin (scCPX; residues 1–134, with D27L, E34F, R37A mutations) into pre-fusion SNAREΔ60 complex describes a multi-site interaction of CPX. The solid lines represent the predicted binding thermogram assuming that both scCPX and truncated SNAREΔ60 are bivalent with well-defined independent thermodynamic parameters describing CPX central helix and CPX accessory helix binding (**B**). (**C**) Representative thermogram of full-length wild-type CPX (residues 1–134) titrated into unblocked SNAREΔ60.

distance in the pre- and post-fusion crystal structures (*Kümmel et al., 2011*). Further, a study by Choi et al. suggests that a completely unstructured CPX would result in a higher FRET signal than a more structured form (*Choi et al., 2011*), so the decrease in FRET efficiency for CPX bound to SNAREΔ60 is not consistent with increased disorder. (2) For both the FRET pairs tested, acceptor placed at residue 31 on $CPX_{acc}$ shows weaker FRET compared with residue 38, consistent with the idea that the $CPX_{acc}$ extends away from the pre-fusion SNARE complex, with dye on residue 31 locating further away from the donor (on SNAP25) than the dye on residue 38 (*Kümmel et al., 2011*, *Figure 4*).

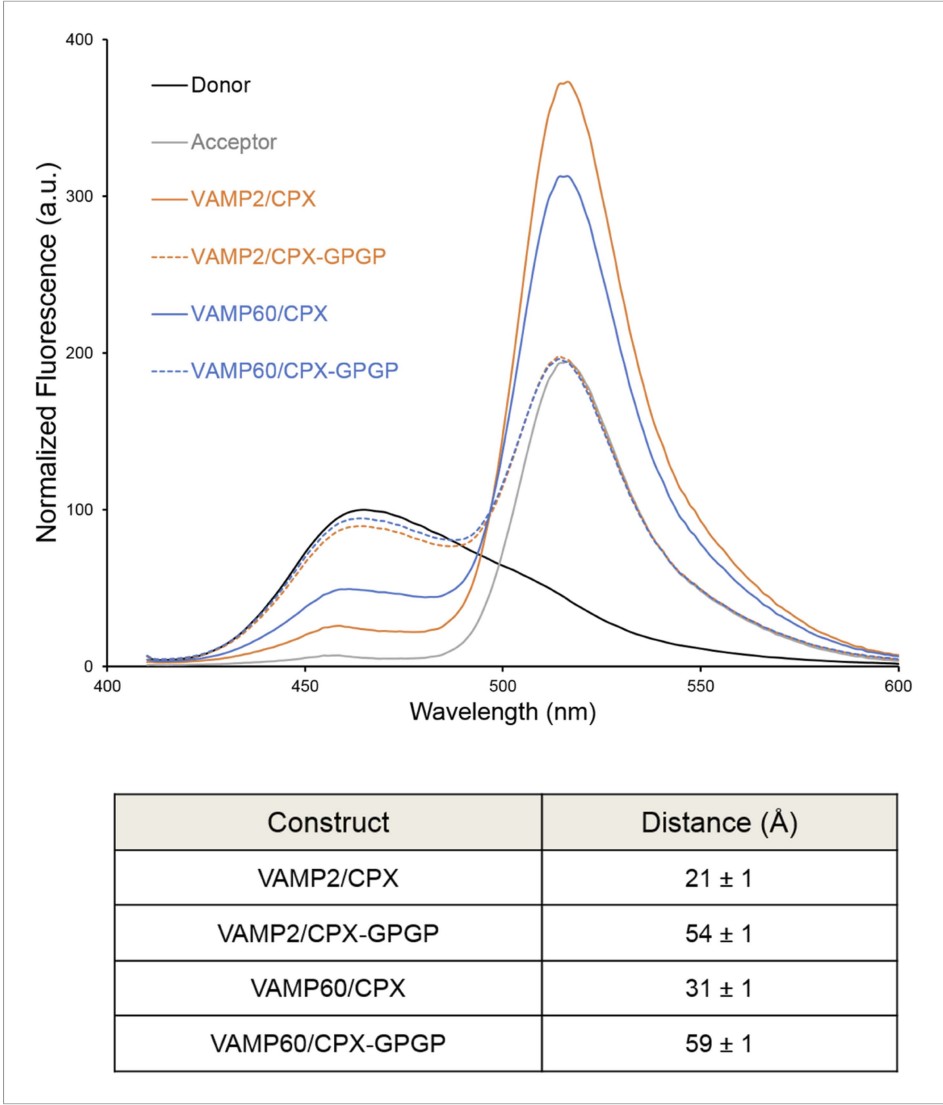

| Construct | Distance (Å) |
|---|---|
| VAMP2/CPX | 21 ± 1 |
| VAMP2/CPX-GPGP | 54 ± 1 |
| VAMP60/CPX | 31 ± 1 |
| VAMP60/CPX-GPGP | 59 ± 1 |

**Figure 4**. Bimane–Oregon green fluorescence resonance energy transfer (FRET) experiments probing the effect of flexibility on complexin (CPX) orientation in pre- and post-fusion CPX–SNARE complexes. The FRET labeling positions were residue 193 on SNAP25 (Bimane) and residue 38 on CPX (Oregon green). Fluorescence emission spectra of Bimane only (black), Oregon green only (grey), and Bimane–Oregon green labeled CPX–SNARE complexes containing VAMP2 (residues 25–96, orange), VAMP60 (residues 25–60, blue) confirms the angled (low FRET) configuration of CPX accessory helix ($CPX_{acc}$) in the pre-fusion SNARE complex (VAMP60). The near complete loss of FRET for the flexible CPX mutant (helix breaking GPGP insertion between $CPX_{acc}$ and CPC central helix ($CPX_{cen}$); CPX–GPGP, dashed lines) compared with the wild-type (WT, solid lines) in both pre- and post-fusion SNARE complexes shows that the difference in FRET signal observed with WT CPX is not due to increased flexibility of the $CPX_{acc}$ in the pre-fusion complex. FRET distances in these CPX–SNARE complexes determined from the quenching of the donor (Bimane) fluorescence is shown (table) and standard deviations are reported from 2–3 independent experiments.

(3) In the accompanying paper (*Krishnakumar et al., 2011*), we tested the effect of VAMP truncation on the orientation of $CPX_{acc}$ and observed the same well-defined low FRET state (∼15% FRET) for CPX bound to SNAREΔ69, SNAREΔ65, or SNAREΔ60 complexes (*Krishnakumar et al., 2011*, *Figure 1B*). But NMR analysis for same or similarly truncated SNARE complexes showed equally dramatic enhancement of local flexibility in $CPX_{acc}$ with increasing truncation on the VAMP c-terminus (SNAREΔ68 versus SNAREΔ62 versus SNAREΔ60 in *Trimbuch et al. (2014)*, *Figure 2C,F*). (4) Finally, the FRET signal we observed in the 'low' FRET state is higher than the FRET signal obtained when we

used a CPX construct with enhanced flexibility (CPX–GPGP) in which a helix-breaking GPGP linker was inserted between the central and accessory helices of CPX for either SNAREΔ60 or the post-fusion SNARE complex (*Kümmel et al., 2011*, *Figure 4D*). Trimbuch et al. argued that, since we observe only a small change in FRET signal for CPX–GPGP compared with CPX–SNAREΔ60, the results obtained with the GPGP mutant are not conclusive. To address this concern we tested CPX–GPGP bound to SNAREΔ60 and SNARE complex using the medium range Bimane/Oregon green FRET pair ($R_0 \sim 38$Å). As shown in *Figure 4*, the GPGP insertion results in a dramatic decrease in the FRET signal, with the $CPX_{acc}$ locating further away from both SNAREΔ60 and the SNARE complex. These data suggest that the increased flexibility does not explain the decreased FRET signal in the pre-fusion truncated SNARE complex.

Thus, taken together, our data are consistent with our earlier conclusion that the low FRET state corresponds to a defined angled configuration on the $CPX_{acc}$ in the pre-fusion complex. This is corroborated by functional data showing that the inhibitory function of CPX requires a fully-folded and rigid CPX helix (*Radoff et al., 2014*), and introducing flexibility within the minimal functional domain of CPX (residue 26–83) results in a reduction or loss of clamping ability corresponding to the degree of instability introduced (*Kümmel et al., 2011*; *Cho et al., 2014*; *Radoff et al., 2014*).

## NMR analysis

Using NMR, Trimbuch et al. were unable to detect any interaction between a peptide corresponding to $CPX_{acc}$ only (CPX26–48) and SNAREΔ60 (*Trimbuch et al., 2014*). This experiment is in agreement with our ITC measurements, which did not detect an interaction between SNAREΔ60 and CPX26–48 alone (unpublished). We suspect that the conformations sampled by CPX26–48 are dependent on its protein context, and this might account for the different binding affinities observed for the shorter (CPX26–48) compared with the longer (CPX26–83 and CPX1–134) constructs. For example, our circular dichroism (CD) measurements showed that $CPX_{acc}$ (residues 26–48) by itself is unfolded, but CPX26–83 forms a stable α-helical structure (*Figure 5A*).

However, *Trimbuch et al. (2014)* also did not observe any dramatic shifts and/or broadening of the $CPX_{acc}$ cross-peaks, as would be expected for insertion into the t-SNARE groove (*Kümmel et al., 2011*), when they used CPX26–83. We note that the prediction from our model is that the *trans* $CPX_{acc}$–t-SNARE interactions would lead to the formation of CPX/SNAREΔ60 polymers, which would not be visible in NMR studies due to line broadening. Thus, if there is a *trans* interaction, then the NMR technique may not be well suited for its interrogation. This cross-linked high-order oligomeric state is evident qualitatively in the ITC experiments where we titrated full-length CPX into unblocked SNAREΔ60 (*Figure 3*). Dynamic light scattering (DLS) analysis (*Figure 5B*, black circles) similarly suggests that CPX26–83/SNAREΔ60 complexes begin to oligomerize at the concentration regime used in Trimbuch et al. (25–50 μM). The DLS data are consistent with a model predicting the size of the oligomers (see 'Materials and methods' for details on the model, *Figure 5B*, black line) which incorporates the $K_d$ of ~25 μM obtained from our ITC analysis of the CPX26–83 and blocked SNAREΔ60 interaction. In contrast, we observe only a small change in the apparent size if we disrupt the binding of $CPX_{acc}$ to t-SNARE either by introducing non-clamping mutations (ncCPX26–83; A30E, A31E, L41E, A44E) in $CPX_{acc}$ (*Figure 5B*, blue circles) or by blocking the t-SNARE groove using a fully assembled ternary SNARE complex (*Figure 5B*, green circles). Based on our oligomerization model, this small change in the particle size could only be fitted by assuming a very low affinity interaction ($K_d \geq 250$ μM) such as might arise from non-specific aggregation (*Figure 5B*, blue and green lines).

We suspect that the concentrations used in the NMR studies (~25–50 μM) (*Trimbuch et al., 2014*), which correspond to the lower end of the range of concentration used for NMR studies, were adjusted to avoid oligomerization and line broadening effects. Because small proteins are difficult to quantitate and because our $K_d$ values, which depend on concentration measurements for their accuracy, are only approximate, it is plausible that, under the NMR conditions of Trimbuch et al., most of the CPX–SNAREΔ60 complexes were monomeric, with only a small fraction in an oligomeric form. Thus, under the experimental conditions used by Trimbuch et al., $CPX_{acc}$ might not have an appreciable interaction with the t-SNARE groove, and the minor bound fraction would not be visible in NMR spectra owing to the large size of the cross-linked oligomers, assuming that there is no exchange between the monomeric and oligomeric forms of CPX–SNAREΔ60. Therefore, we believe that the negative NMR data in Trimbuch et al. do not preclude a *trans* interaction between CPX and the SNAREΔ60 complex.

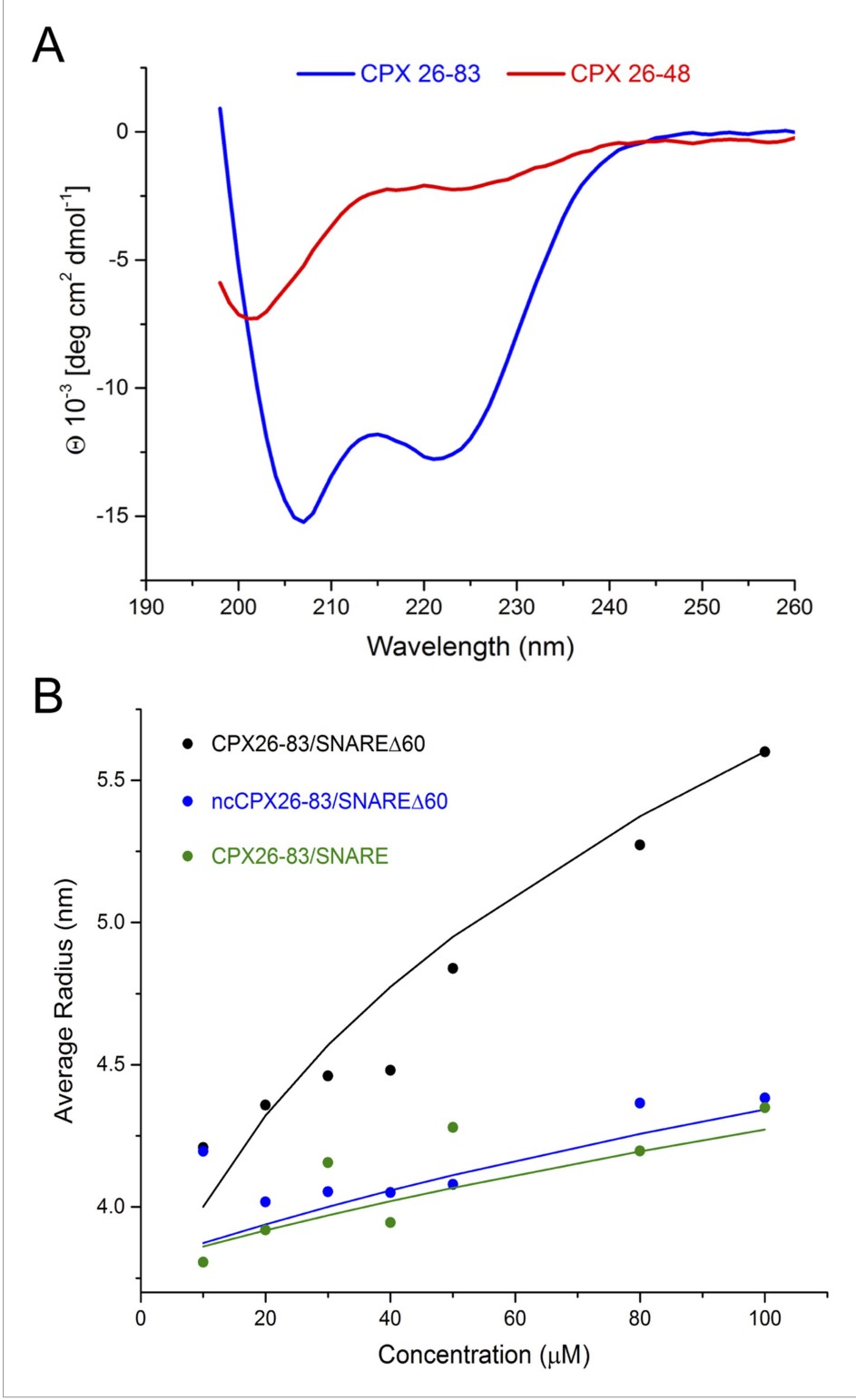

**Figure 5**. Circular dichroism (CD) and dynamic light scattering (DLS) experiments. (**A**) CD spectra of the complexin (CPX) accessory domain (CPX26–48) and the CPX minimal functional domain (CPX26–83). The continuous CPX construct (26–83, blue) with uninterrupted accessory and central domain shows the characteristics of an α-helical protein. In contrast, the isolated CPX accessory domain (26–48) shows little secondary structure and appears mostly unfolded. This may explain why no interaction between the CPX accessory region and SNAREΔ60 was observed by NMR. (**B**) DLS analysis showing the formation of the high-order oligomers of CPX–SNAREΔ60 in the concentration

*Figure 5. continued on next page*

*Figure 5. Continued*

range used in the NMR experiments in *Trimbuch et al. (2014)*. Experimental average particle radius of pre-formed CPX26–83/SNAREΔ60 (black dots), ncCPX26–83/SNAREΔ60 (blue dots), and CPX26–83/SNARE (green dots) at varying concentration is shown. The solid lines (same color scheme) represent the average gyration radius of the oligomers calculated from the semi-quantitative model described in 'Materials and methods'.

## In vivo analysis

In Trimbuch et al., the authors tested the $CPX_{acc}$/t-SNARE insertion model using the autaptic neuronal culture derived from CPX triple KO mice (CPX TKO). They found that the super clamp and non-clamp CPX mutations, which are predicted to enhance or decrease $CPX_{acc}$–t-SNARE binding, respectively, did not have the expected effect on evoked or spontaneous release. Based on this, they concluded that functional data do not support the $CPX_{acc}$ insertion/zig-zag model (*Trimbuch et al., 2014*) and raised concerns regarding the use of the cell–cell fusion assay (*Giraudo et al., 2006*, *2009*; *Krishnakumar et al., 2011*; *Kümmel et al., 2011*) as an in vitro system to study CPX clamping.

These results and conclusions must be viewed with caution since the studies in knockout neurons, particularly in autaptic neuronal cultures, reveal a more restricted role of CPX in the neurotransmitter release (*Reim et al., 2001*; *Xue et al., 2007*, *2008*; *Yang et al., 2013*). Specifically, the inhibitory function of CPX has not been observed in the autaptic system: that is, both evoked and spontaneous release are reduced in CPX TKO mice (*Reim et al., 2001*; *Xue et al., 2008*; *Trimbuch et al., 2014*). In contrast, other preparations like rodent mass cultured neurons with reduced CPX expression and invertebrates lacking CPX exhibit reduced evoked release and enhanced spontaneous release (*Huntwork and Littleton, 2007*; *Yang et al., 2010*; *Cho et al., 2014*). Therefore, these systems may be more relevant to examine the mechanism of CPX function in regulating vesicle fusion, particularly in its ability to inhibit and regulate spontaneous release. In rescue experiments with rodent mass cultured neurons in which CPX was knocked down, CPX mutants that either enhanced or disrupted the $CPX_{acc}$–SNARE interactions (super-clamp and poor clamp, respectively) resulted in corresponding reductions and enhancements in spontaneous release compared with wild-type CPX controls (*Yang et al., 2010*). Recent experiments at the *Drosophila* neuromuscular junction, which in the absence of CPX exhibits large increases in spontaneous release, directly examined the role of a CPX zig-zag array in regulating spontaneous release. Using genetic rescue approaches, CPX mutants predicted to prevent the formation of the zig-zag array (CPX–GPGP) disrupt the ability of CPX to clamp spontaneous release, while mutations predicted to enhance the $CPX_{acc}$–t-SNARE interaction (super-clamp mutation) exhibit a strong clamping ability (*Cho et al., 2014*). These results taken together are consistent with a CPX cross-linking model mediating the CPX clamping function to regulate spontaneous release, even though recent reports suggest that the inhibitory and activating functions of CPX may be separable (*Yang et al., 2010*; *Cho et al., 2014*; *Lai et al., 2014*). In addition, these in vivo studies (*Yang et al., 2010*; *Cho et al., 2014*) support many of the observations made first with the cell–cell fusion assay (*Giraudo et al., 2006*, *2008*, *2009*; *Krishnakumar et al., 2011*; *Kümmel et al., 2011*; *Li et al., 2011*), underscoring the validity and the relevancy of the in vitro cell–cell fusion assay to study CPX clamping.

## Electrostatic hindrance model

As an alternative to the insertion/zig-zag model, Trimbuch et al. advanced an 'electrostatic hindrance model' for CPX clamping (*Trimbuch et al., 2014*). This was based on the finding that increasing or decreasing the net negative charge on $CPX_{acc}$ inhibits or stimulates neurotransmitter release, respectively (*Trimbuch et al., 2014*). We note that a number of both in vitro and in vivo studies on $CPX_{acc}$ modification argue strongly against the electrostatic hindrance model (*Yang et al., 2010*; *Kümmel et al., 2011*; *Cho et al., 2014*; *Radoff et al., 2014*). Introducing two (L41E, A44E) or four (A30E, A31E, L41E, A44E) negatively charged mutations in the mammalian $CPX_{acc}$ has been shown to entirely abolish or severely diminish the interaction of $CPX_{acc}$ with the mammalian or *Drosophila* t-SNARE, respectively (*Kümmel et al., 2011*; *Cho et al., 2014*). Consistent with this finding, the similar mutations have been shown to abrogate the clamping function of CPX in both in vitro cell–cell fusion assay (*Kümmel et al., 2011*) and in vivo rescue experiments with knockdown neurons (*Yang et al., 2010*). Further, in contrast to the electrostatic model, increasing the hydrophobicity of $CPX_{acc}$ has been shown to enhance the clamping function of CPX (*Yang et al., 2010*; *Kümmel et al., 2011*; *Cho et al., 2014*).

## Discussion

We believe that the new in vitro data presented above and physiological data published recently (*Cho et al., 2014*) should dispel the concerns raised by Trimbuch et al. regarding the plausibility of the insertion/zig-zag model. Specifically, ITC binding experiments with blocked SNARE complexes, where a 2.5–3-fold excess of blocking peptide was present, have corroborated the interaction of CPX$_{acc}$ with the t-SNARE groove in a pre-fusion (truncated) SNARE complex as observed in the crystal structure (*Figure 2* and *Kümmel et al., 2011*). Additionally, titration of scCPX into unblocked truncated SNARE complex revealed multiple binding sites consistent with independent binding sites for CPX$_{cen}$ and scCPX$_{acc}$, thus describing the essential feature of the *trans* clamping interaction of CPX (*Figure 3*). FRET analysis (using two FRET pairs with different R$_0$ values) with 'flexible' CPX construct (CPX–GPGP) has clearly demonstrated that the low FRET state corresponding to the angled conformation, which describes the *trans* clamping interaction, is due to discrete conformational change and not because of increased CPX flexibility (*Figure 4* and *Kümmel et al., 2011*). Lastly, the physiological relevance of the insertion model was established by independent genetic rescue experiments in the *Drosophila* neuromuscular junction, wherein mutations in CPX that are predicted to prevent the formation of the SNAREpin array were found to disrupt the ability of CPX to clamp spontaneous release (*Cho et al., 2014*).

All of our data are consistent with a model in which CPX$_{acc}$ from one SNAREpin interacts in *trans* with the t-SNARE groove of a second SNAREpin. As we emphasized in Kümmel et al., we do not propose that the interactions are necessarily exactly as observed in a single crystal structure for which scCPX was used and where hydrophobic accessory helix residues (which are hydrophilic in wild-type CPX) interact with hydrophobic residues in a partially assembled SNARE complex. Details of the interaction between wild-type CPX$_{acc}$ and t-SNARE must be different (they are also weaker, explaining why the wild-type CPX clamps less well), and we can well imagine a scenario in which wild-type CPX$_{acc}$ interacts with a slightly different surface of the assembling SNARE complex. In fact, we found that a single mutation in the CPX$_{acc}$ (F34M) results in two distinct binding interfaces for scCPX, but both giving rise to the same zig-zag topology. Both interactions involved the same face of CPX$_{acc}$ and t-SNARE, although the binding site on t-SNARE was extended by two helical turns for the mutant (*Kümmel et al., 2011*). The consideration that interactions of wild-type and scCPX and the SNARE complex are not identical does not, however, exclude the possibility that a *trans* interaction of some sort is responsible for clamping. We also suspect that yet unknown additional interactions in the pre-fusion complex, which are not represented in our crystal structure, may further stabilize the *trans* clamping interaction (*Cho et al., 2014*; *Radoff et al., 2014*). We do not consider the *trans* clamping model proven, but it remains the most well-defined model for which a good amount of evidence is available.

## Materials and methods

### Plasmid constructs and protein purification

The constructs used in this study are GST-PreScission-VAMP2Δ60 (human VAMP2 residues 29–60), pET15b-oligohistidine-thrombin-VAMP2 (human VAMP2 residues 29–96), GST-TEV-syntaxin1A (containing rat syntaxin 1a residues 191–253), oligohistidine-MBP-thrombin-SNAP25N (containing human SNAP25A residues 7–82 and a C-terminal tryptophan), GST-TEV-SNAP25C (containing human SNAP25A residues 141–203), and GST-PreScission-CPX containing human complexin1 residues 1–134 (CPX); residues 1–134 with super-clamp mutations D27L, E34F, R37A (scCPX); residues 1–134 with flexible GPGP insert between residues 49–50 (CPX–GPGP), residues 26–83 (CPX26–83); residues 26–83 with non-clamping mutations A30E, A31E, L41E, A44E (ncCPX26–83) residues 48–134 (CPX-48) and residues 26–48 (CPX26–48). All constructs were expressed and purified as described previously (*Krishnakumar et al., 2011*; *Kümmel et al., 2011*). To ensure high quality, all proteins were purified on a High-load Superdex 75 (16/60, GE Healthcare; Piscataway, NJ) gel filtration column.

### ITC analysis

ITC experiments were carried out as described previously (*Krishnakumar et al., 2011*; *Kümmel et al., 2011*; *Cho et al., 2014*). To assemble the post-fusion and pre-fusion SNAREΔ60 complex, Syntaxin 1a, SNAP25N, SNAP25C, and VAMP2 or VAMPΔ60 were mixed (molar ratio of 1:1.2:1.2:1.2 for the post-fusion complex and 1:1.2:1.2:1.6 for pre-fusion SNAREΔ60 complex) and incubated overnight at

4°C. The assembled complexes were purified from non-productive aggregates and unassembled components by gel filtration (High-load Superdex 75 16/60, GE Healthcare). To form the blocked SNAREΔ60 complex, purified SNAREΔ60 complex was mixed with 2.5 molar excess of CPX-48 and incubated overnight at 4°C to ensure complete binding. To ensure buffer uniformity to measure the weak interactions, CPX variants and the different SNARE complexes were extensively dialyzed (4 L for 4 hr followed by another 4 L overnight) into the same phosphate buffered saline (PBS) buffer (pH 7.4, 137 mM NaCl, 3 mM KCl, 10 mM sodium phosphate dibasic, 2 mM potassium phosphate monobasic, 0.25 mM TCEP) before the ITC analysis. The concentrations of dialyzed proteins were determined by bicinchoninic acid (BCA) protein assay kit (Thermo Scientific; Waltham, Ma) and/or Bradford assay (Bio Rad; Hercules, CA) with bovine serum albumin (BSA) as the standard. ITC experiments were performed with a Microcal ITC200 (Malvern Instruments, UK). Typically, ~200 µl of SNARE solution was loaded into the sample cell and ~60 µl of CPX solution was loaded into the syringe. The protein concentrations used for the titration were as follows: 110 µM CPX-48 titrated into 5.8 µM SNARE (*Figure 1A*); 210 µM CPX-48 titrated into 14 µM SNAREΔ60; ~360 µM CPX1–134 or CPX26–83 titrated into ~20 µM blocked SNAREΔ60 (*Figure 2*); 150 µM wild-type or scCPX1–134 titrated into 7.5 µM SNAREΔ60 (*Figure 3*). The heat change from each injection was integrated and then normalized by the moles of CPX in the injection. The thermographs were analyzed by non-linear least squares fit with the one-set-of-sites-model in Microcal Origin ITC200 software package to obtain the stoichiometric number (N), the molar binding enthalpy (ΔH), and the association constant ($K_a$). The equilibrium dissociation constant ($K_d$), the binding free energy (ΔG), and the binding entropy (ΔS) were calculated using the thermodynamic equations:

$$K_d = {}^1/_{K_a}.$$

$$\Delta G = \Delta H - T\Delta S = -RT\ln(K_a).$$

## FRET analysis

FRET measurements were carried out as described previously (*Krishnakumar et al., 2011*; *Kümmel et al., 2011*). SNAP25 D193C was labeled with the donor probe, Bimane (Monochlorobimane, Invitrogen) and the scCPX Q38C was labeled with acceptor Oregon green (Oregon green 488 maleimide, Invitrogen). The proteins were labeled using 10× molar excess of dye in 50 mM Tris buffer, pH 7.4, containing 150 mM NaCl, 10% glycerol, and 1 mM TCEP. Following overnight incubation at 4°C, the excess dye was separated from the labeled proteins using a NAP desalting column (GE Healthcare). The labeling efficiency was calculated using $\varepsilon_{396} = 5300$ L M$^{-1}$ cm$^{-1}$ for Bimane and $\varepsilon_{496} = 76,000$ L M$^{-1}$ cm$^{-1}$ for Oregon green, and the protein concentration was measured using the Bradford assay with BSA as standard. Typically, the labeling efficiency was >95% for both Bimane-SNAP25 and Oregon green-CPX. The double-labeled CPX–SNARE complexes were assembled overnight at 4°C and purified by gel filtration on a Superdex 75 (10/30, GE Healthcare) gel filtration column. All fluorescence data were obtained on a Perkin-Elmer (Waltham, MA) LS55 luminescence spectrometer operating at 25°C. Excitation and emission slits of 5 nm were used in all measurements. Fluorescence emission spectra were measured over the range of 410–600 nm with the excitation wavelength set at 396 nm. The donor probe concentration was adjusted to 2 µM in all samples.

## Circular dichroism (CD) analysis

CD spectra of peptides corresponding to the CPX accessory domain alone (residues 26–48) or the minimal functional domain (residues 26–83) were recorded in PBS using Chirascan CD spectrometer (Applied Photophysics, UK) at 25°C from the range of 260 nm–198 nm at 1 nm bandwidth.

## Dynamic light scattering (DLS) analysis

CPX–SNAREΔ60 complexes containing wild-type CPX1–134 were assembled and purified as previously described (*Krishnakumar et al., 2011*; *Kümmel et al., 2011*). DLS experiments were carried out on a DynaPro NanoStar instrument (Wyatt Technology; Santa Barbara, CA) at a wavelength of 663.76 nm operating at 4°C. Protein samples were centrifuged (10 min at 13,000×g) and data were collected using DynaPro disposable cuvettes. Autocorrelations for 20 s were collected over 15 acquisitions. Points were eliminated if the intensity fluctuated by more than 15% from the average. Data were analyzed with DYNAMICS 7.1.7.16 software (Wyatt Technology).

In DLS there is oligomerization of CPX–SNAREΔ60. We can try to model this process as an equilibrium between bound and unbound CPX–SNAREΔ60, wherein the bound state is generated by the interaction of $CPX_{acc}$ with the t-SNARE groove. Thus, this equilibrium can be written as:

$$CPX - SNARE\Delta60 \ + \ CPX - SNARE\Delta60 \overset{K_d}{\leftrightarrows} CPX - SNARE\Delta60 - CPX - SNARE\Delta60. \tag{1}$$

which, in terms of concentration, gives:

$$[un - bound \ CPX - SNARE\Delta60]^2 \ = \ Kd * [bound \ CPX - SNARE\Delta60]. \tag{2}$$

Keeping in mind that

$$[un - bound \ CPX - SNARE\Delta60] \ + \ [bound \ CPX - SNARE\Delta60] \ = \ [CPX - SNARE\Delta60]_{initial}, \tag{3}$$

where $[CPX–SNARE\Delta60]_{initial}$ is the initial concentration, the equilibrium concentrations can be calculated from (2) and (3) as:

$$un - bound \ CPX - SNARE\Delta60] \ = \ \frac{K_d\left(\sqrt{1 + \frac{4[CPX - SNARE\Delta60]_{initial}}{K_d}} - 1\right)}{2}. \tag{4}$$

$$[bound \ CPX - SNARE\Delta60] \ = \ [CPX - SNARE\Delta60]_{initial} - \frac{K_d\left(\sqrt{1 + \frac{4[CPX - SNARE\Delta60]_{initial}}{K_d}} - 1\right)}{2}. \tag{5}$$

Then, the probability that an accessory helix is bound is given by:

$$p = \frac{[bound \ CPX - SNARE\Delta60]}{[CPX - SNARE\Delta60]_{initial}} = 1 - \frac{K_d\left(\sqrt{1 + \frac{4[CPX - SNARE\Delta60]_{initial}}{K_d}} - 1\right)}{2[CPX - SNARE\Delta60]_{initial}}, \tag{6}$$

and the average number of monomers is equal to:

$$N = \frac{p}{1 - p} + 1 = \frac{2[CPX - SNARE\Delta60]_{init}}{K_d\left(\sqrt{1 + \frac{4[CPX - SNARE\Delta60]_{init}}{K_d}} - 1\right)}. \tag{7}$$

We will model the gyration radius of an N-mer as:

$$r \ = \ N^{0.5} \ r_0, \tag{8}$$

where $r_0$ is the distance between two particles in the array and the reflection of monomer size. The r value is quantitative for long ideal polymers but is only semi-quantitative for non-ideal oligomers under consideration here.

The experimental results can be well approximated by *Equation 8* using $r_0 = 3.5$ nm and the $CPX_{acc}$–t-SNARE interaction described by $K_d = 25$ μM for CPX26–83 titrated into blocked SNAREΔ60. This $r_0$ value is reasonable considering the dimensions of the CPX–SNAREΔ60 unit within the zig-zag array. Hence, the oligomerization we observe by DLS is consistent with the $K_d$ values we measured by ITC (*Figure 2*). In support of the idea that the $CPX_{acc}$–t-SNARE clamping interaction results in the oligomeric state, CPX–SNARE complexes containing either ncCPX26–83 (A30E, A31E, L41E, A44E) or the full ternary SNARE complex show only a small change in the average particle radius. This behavior can be modeled using *Equation 8* with $r_0 = 3.8$ nm and only if we assume a very low affinity interaction, namely $K_d = 250$ μM for ncCPX into SNAREΔ60 and $K_d = 300$ μM for CPX into the full ternary SNARE complex. This suggests that this change in particle size might be a result of non-specific aggregation. Note that $r_0$ values for specific and non-specific oligomerization are not necessarily expected to be identical.

## Acknowledgements

We thank Dr Richard Cho for input and careful reading of the manuscript. This work was supported by grants from the National Institute of Health (NIH) to JER (GM 071458) and KMR (GM 080616).

## Additional information

### Funding

| Funder | Grant reference | Author |
| --- | --- | --- |
| National Institutes of Health (NIH) | GM 071458 | James E Rothman |
| National Institutes of Health (NIH) | GM 080616 | Karin M Reinisch |

The funder had no role in study design, data collection and interpretation, or the decision to submit the work for publication.

### Author contributions

SSK, FL, DK, Conception and design, Acquisition of data, Analysis and interpretation of data, Drafting or revising the article; JC, Acquisition of data, Contributed unpublished essential data or reagents; CMS, Acquisition of data, Analysis and interpretation of data; FP, JER, KMR, Conception and design, Analysis and interpretation of data, Drafting or revising the article

### Author ORCIDs

Daniel Kümmel, http://orcid.org/0000-0003-3950-5914

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
