## [Decision Letter]

Thank you for sending your work entitled “Re-visiting the *trans* Insertion Model for Complexin Clamping” for consideration at *eLife*. Your article has been favorably evaluated by Randy Schekman (Senior editor) and 3 reviewers, one of whom is a member of our Board of Reviewing Editors.

The following individuals responsible for the peer review of your submission have agreed to reveal their identity: Axel Brunger (Reviewing editor) and Ad Bax (Peer Reviewer). A third reviewer remains anonymous.

The Reviewing editor and the other reviewers discussed their comments before we reached this decision, and the Reviewing editor has assembled the following comments to help you prepare a revised submission.

In this article the authors present arguments to dispute the findings and conclusions drawn by Trimbuch et al. (*eLife* 2014). Most importantly, the authors measure a ten times higher affinity between the complexin accessory helix and blocked SNARE complex by ITC, and another ITC experiment provides evidence for multiple binding sites of complexin to SNARE complex. The authors speculate that the higher affinity is due to better homogeneity of their purified SNARE complexes, and argue that lower affinity (presumably caused by heterogeneity) of the Trimbuch et al. complexes combined with the high mobility of the side-chains (as determined by the Kümmel et al. crystal structure) would explain why Trimbuch et al. could not detect an interaction between the complexin accessory domain and SNARE complex by ITC and NMR. The authors also provide arguments that the neuronal culture by Trimbuch et al. may represent a special case, and that the electrostatics hindrance model is in disagreement with other data.

We note that simply mixing syntaxin, S25 and synaptobrevin/VAMP is well-known to generate all sorts of complexes other than the canonical SNAREs (examples include 2:1 Sx:S25 or antiparallel SNAREs as in Weninger, PNAS 2004). However, the NMR spectra by Trimbuch et al. appear to argue against this possibility (see point 2 below). The increased ITC affinity data for binding of complexin to the truncated SNARE complex compared to the data by Trimbuch et al. is compelling. However, it is not clear how to explain the lack of interactions as observed by the NMR experiments by Trimbuch et al.

Major comments:

1) The authors should confirm that it is indeed sample homogeneity that explains the difference in affinities measured by ITC with apparently identical (or very similar) SNARE and complexin constructs. Ideally, complexes should be purified according to the protocol by Trimbuch et al. and by the protocol described in this paper, and then ITC experiments performed side-by-side with both sets of proteins. We note that other potential differences could be: presence or absence of tags (apparently all tags in the present work have been removed, but there is no information in the paper by Trimbuch et al), the concentration used in the various experiments (there is no information about the protein concentration used for the ITC experiments in the present work), the buffer conditions, and the possibility of oligomers of the various SNARE complexes.

2) The criticisms of Trimbuch's NMR results are problematic, as that NMR data appears fairly solid. For example, the statement that “the NMR spectra similarly speak to this flexibility as opposed to complete unfolding”, for example, is not warranted. If the backbone were even somewhat ordered, one would expect substantial chemical shift changes upon interaction, as well as broadening due to the increased spectral density at zero frequency (J(0)), even if the backbone order parameter, S^2, were as low as, say, 0.5. The fact that CPX26-48 shows considerably less helicity (by CD) than CPX(26-83) (Figure 5) is a valid criticism, as apparently a high degree of alpha helicity of residues 30-48 requires the presence of residues 49-65. However, considering that the accessory helix is clearly in a dynamic equilibrium between transient helix and coil, and this equilibrium is shifted from perhaps 60% helicity (based on 13Ca shifts in Pabst, JBC 2000, to perhaps ∼10-20% (?) based on the CD data of Figure 5) one would still expect to see this interaction as NMR is incredibly sensitive to even very weak binding. So, even while shortening the accessory helix indeed is expected to weaken the affinity, it would not explain the complete absence of an interaction, seen in Figure 2 of Trimbuch or even in the superclamp CPX Figure 2 spectrum of their supplement 2. Even if the affinity were as weak as ∼1 mM, one would still expect to see the onset of peak shifting and broadening in such spectra.

In the subsection headed “NMR Analysis”, the authors attribute the absence of evidence for an interaction between the CPX accessory helix and SNAREdelta60 again to the heterogeneous nature of the truncated SNARE complexes. The NMR spectra presented in Figure 3 of Trimbuch et al. of the full-length and truncated SNARE complex show no sign of heterogeneity of the SNARE complexes themselves. Morever, Trimbuch et al. also do not see any evidence for such an interaction in SCdelta(68) or for the full length complex, and not even for the superclamp-Cpx, whereas comparison of the interactions/shift changes observed for the central helix point to a tight complex for full length SC, with progressively less tight (fast exchange) binding for CPX residues in the “linker region” of complexin (residues 58-69). Notably, more C-terminal CPX resonances (73, 80, 82, 83) are completely unaffected by SCdelta68 and SCdelta62 mutations, and assuming these residues make authentic interactions in the full length complex, these still must be present in the truncated complexes. So, the fact that Trimbuch's central helix residues adopt unique chemical shifts with rather uniform line widths effectively rules out the possibility of a heterogeneous (truncated, 2:1 or otherwise) distribution of truncated SNARE complexes. Other explanations must be provided in the present work in order to criticize the NMR data by Trimbuch et al.

3) The criticism in Trimbuch of the original FRET results by Kümmel et al. is that the decreased FRET in the cases interpreted by Krishnakumar as indicating the complexin accessory helix angling away from the SNARE bundle could also be interpreted as increased flexibility in the structured state due to the mutations and truncations. The response of the current paper is to include new data with a FRET pair having a longer Rzero, showing more substantial elimination of FRET due to the GPGP insertion, for which the small but possibly detectable FRET change in the previous experiments with the shorter Rzero dye pair. Despite the fact that the new longer-Rzero, GPGP-insert data support Kümmel's interpretation that complexin angles away modestly, not totally flexible, the reviewers don't view these FRET experiment as conclusive evidence of either static or dynamic configurations. They don't have the distance or time resolution needed. Rather, they are useful observations that can be consistent with models derived from aggregating several other experiments.

In this context, the authors may wish to consult the paper by Choi et al, Structure, 2011 (10.1016/j.str.2011.01.011) that showed that complexin is in an extended conformation when bound to complete SNARE complex, whereas it is in a more coiled conformation when unbound. Thus, if the unbound part of complexin were completely unstructured when bound to the truncated SNARE complex, one would expect actually higher FRET efficiency to the label site on the SNARE complex. In contrast, a lower FRET efficiency is observed, consistent with the notion that complexin is in an angled conformation when bound to truncated SNARE complex. However, the mutation shown in Figure 4 induces a further decrease of FRET efficiency, a result that is incompatible if complexin were unstructured (in the subsection headed “FRET Analysis”). Rather, this insertion probably introduces a hinge between the core and accessory domains, which may explain the decreased FRET efficiency. A discussion of these points should be included.

[Editors' note: further revisions were requested prior to acceptance, as described below.]

Thank you for sending your work entitled “Re-visiting the *trans* Insertion Model for Complexin Clamping” for consideration at *eLife*. Your article has been favorably evaluated by Randy Schekman (Senior editor), a Reviewing editor, and 2 reviewers.

The Reviewing editor and the other reviewers discussed their comments before we reached this decision, and the Reviewing editor has assembled the following comments to help you prepare a revised submission.

We thank the authors for addressing many of our concerns and the inclusion of new supporting data (DLS experiment). Overall, the difference in ITC measurements reported in this work and the work by Trimbuch et al. remains the most compelling evidence that warrants publication of this work. We encourage communication between the two groups to resolve this issue after this work has been accepted.

The revised manuscript requires some revision. In particular, we strongly encourage control experiments for the newly presented DLS data. We promise expeditious consideration after the revision has been received and anticipate full acceptance assuming that the suggested control experiments produce the expected outcome.

Comments:

1) The authors now show new and highly relevant DLS data, indicating that indeed translational diffusion decreases upon increasing the concentration of CPX-SNARE_delta60 (please clarify whether this DLS data is indeed for CPX or CPX26-83, used for panel A in this figure). In particular if the DLS data was obtained for CPX (Kd ∼15 microM), the DLS data are not particularly compelling as evidence for hetero-oligomerization involving multiple SNARE complexes and complexins, as one would have expected a substantially larger increase in apparent particle radius than observed. Thus, the authors are requested to present a model that might explain their observation.

2) If such a model can be obtained, an important control for the DLS experiment is missing: measuring the DLS signal for a tuncated CPX without the accessory helix in complex with SNARE_delta60. Ideally, the authors should also perform the experiment for the “poor-clamp” mutant (A30E, A31E, L41E, A44E).

3) The very low ellipticity of ∼-7.3*10^-3 at ∼200 nm seen in the CD spectrum of CPX26-48 (Figure 5) is a concern. For example, synaptobrevin produces a minimum of ∼ -13 *10^-3 at ∼200 nm (J.Biol.Chem, 272, 28036-28041, 1997). The concentration measurement of CPX26-48 and normalization of the CD should be checked.

4) Currently, the substantial number of Kd values and their corresponding proteins are distributed throughout the text and figures. It would be useful if they could also be compiled in a single table, such that a reader can more easily compare the various values, along with an explanation of the various constructs and SNARE complexes used, including “blocked” SNARE complex.

5) In the subsection headed “NMR Analysis”, CD only defines total helicity of CPX26-83, and one cannot uniquely assign this to 26-48. Moreover, the NMR study of Pabst et al. (JBC 275:19808-18, 2000) provides fairly strong evidence that residues 28-64 in free complexin form one contiguous helix. Thus, the statement should be corrected accordingly.

---

## [Author Response]

*1) The authors should confirm that it is indeed sample homogeneity that explains the difference in affinities measured by ITC with apparently identical (or very similar) SNARE and complexin constructs. Ideally, complexes should be purified according to the protocol by Trimbuch et al. and by the protocol described in this paper, and then ITC experiments performed side-by-side with both sets of proteins. We note that other potential differences could be: presence or absence of tags (apparently all tags in the present work have been removed, but there is no information in the paper by Trimbuch et al), the concentration used in the various experiments (there is no information about the protein concentration used for the ITC experiments in the present work), the buffer conditions, and the possibility of oligomers of the various SNARE complexes*.

We agree with the reviewers that heterogeneity of the SNARE complexes alone does not explain the absence of any interaction between SNAREΔ60 and complexin as observed by NMR, since the spectra attest to the homogeneity of the SNARE preparation. It is possible that they partly contribute to the discrepancy between our ITC data and that of the other group, but we agree with the reviewers that additional experimental factors, like the buffer conditions, purification methodology (i.e. presence or absence of affinity tags), and the method used to determine the protein concentration (Bradford/BCA or A_280_) might contribute to the variability in the Kd values. The latter is of particular significance since concentration determination of small proteins/peptide (<7 kDa) are highly dependent on the method used. However, we are unable to systematically compare the two studies since several important experimental details (for example, whether affinity tags were present or what method was used for protein quantification) are not described in Trimbuch et al. We have now amended and expanded the relevant section in our main text (under the subsection headed “ITC Experiments”) addressing the difference between the two studies. In addition, we have now included explicit details of our experimental conditions for the ITC experiments, including the concentrations used.

*2) The criticisms of Trimbuch's NMR results are problematic, as that NMR data appears fairly solid. For example, the statement that* “*the NMR spectra similarly speak to this flexibility as opposed to complete unfolding*”*, for example, is not warranted. If the backbone were even somewhat ordered, one would expect substantial chemical shift changes upon interaction, as well as broadening due to the increased spectral density at zero frequency (J(0)), even if the backbone order parameter, S^2, were as low as, say, 0.5. The fact that CPX26-48 shows considerably less helicity (by CD) than CPX(26-83) (*Figure 5*) is a valid criticism, as apparently a high degree of alpha helicity of residues 30-48 requires the presence of residues 49-65. However, considering that the accessory helix is clearly in a dynamic equilibrium between transient helix and coil, and this equilibrium is shifted from perhaps 60% helicity (based on 13Ca shifts in Pabst, JBC 2000, to perhaps ∼10-20% (?) based on the CD data of*
Figure 5*) one would still expect to see this interaction as NMR is incredibly sensitive to even very weak binding. So, even while shortening the accessory helix indeed is expected to weaken the affinity, it would not explain the complete absence of an interaction, seen in*
Figure 2
*of Trimbuch or even in the superclamp CPX*
Figure 2
*spectrum of their supplement 2. Even if the affinity were as weak as ∼1 mM, one would still expect to see the onset of peak shifting and broadening in such spectra*.

*In the subsection headed* “*NMR Analysis*”*, the authors attribute the absence of evidence for an interaction between the CPX accessory helix and SNAREdelta60 again to the heterogeneous nature of the truncated SNARE complexes. The NMR spectra presented in*
Figure 3
*of Trimbuch et al. of the full-length and truncated SNARE complex show no sign of heterogeneity of the SNARE complexes themselves. Morever, Trimbuch et al. also do not see any evidence for such an interaction in SCdelta(68) or for the full length complex, and not even for the superclamp-Cpx, whereas comparison of the interactions/shift changes observed for the central helix point to a tight complex for full length SC, with progressively less tight (fast exchange) binding for CPX residues in the* “*linker region*” *of complexin (residues 58-69). Notably, more C-terminal CPX resonances (73, 80, 82, 83) are completely unaffected by SCdelta68 and SCdelta62 mutations, and assuming these residues make authentic interactions in the full length complex, these still must be present in the truncated complexes. So, the fact that Trimbuch's central helix residues adopt unique chemical shifts with rather uniform line widths effectively rules out the possibility of a heterogeneous (truncated, 2:1 or otherwise) distribution of truncated SNARE complexes. Other explanations must be provided in the present work in order to criticize the NMR data by Trimbuch et al*.

The reviewer comments have prompted us to reconsider our explanation for why the NMR data do not show an in-trans interaction between CPXacc/SNARE 60 complexes. Upon further reflection, we note that the prediction from our model is that in-*trans* CPX_acc_-SNAREΔ60 interactions would lead to the formation of CPX-SNARE 60 polymers at the concentration range used by Trimbuch et al. (∼25-50 μM), and these polymers would not be visible in NMR studies due to line broadening. This cross-linked higher-order oligomeric state is evident qualitatively in the ITC data shown (Figure 3). We have now also observed by DLS that CPX/SNARE60 complexes begin to oligomerize at concentrations of 10-35 μM (new data in Figure 5), in the expected range, if there is a in-trans interaction between CPX_acc_ and SNAREΔ60 with K_d_∼25 μM (Figure 2 with CPX26-83,). We suspect that the concentrations used in the Trimbuch et al. NMR studies (25-50 μM), which are at the lower end of the concentration range used for NMR studies, were adjusted to avoid oligomerization and line broadening effects. Because protein concentrations are difficult to measure accurately, especially for small proteins, and our binding constants (K_d_∼25 μM) are calculated based on the protein concentration, it is plausible that at the concentrations used in the NMR experiments, the complexes were mostly monomeric, with only a small fraction in oligomeric form. Especially if there is no exchange between the monomeric and oligomeric forms of CPX-SNAREΔ60 complexes, then the small fraction of CPXacc/SNAREΔ60 in the oligomers, which would be invisible by NMR, would not be evident from the NMR spectra. Our ITC and the NMR data of Trimbuch et al. are therefore reconcilable, and the NMR data do not necessarily preclude the *trans* interaction observed by ITC.

The NMR data that there is no interaction between the peptide corresponding to CPX accessory helix alone (CPX26-48) and SNAREΔ60 agrees with our unpublished findings from ITC, where we also did not detect an interaction. We should have mentioned this in our first version of this manuscript – this is why, for us, these particular NMR results were “not unexpected”. We did, however, discover the CPX_acc_-t-SNARE interaction with full length CPX or CPX26-83 and the SNAREΔ60 by ITC. Although, it is not entirely clear to us why CPX26-48 does not interact with SNAREΔ60, the CPX26-48 construct clearly behaves very differently in solution from the full length CPX protein. We show data that even CPX26-83 has significantly more helical character than the CPX26-48 (Figure 5). This may explain, at least in part, the lack of an interaction that is observed when using CPX26-48. We also note that the longer versions of CPX (residues 1-134) used in our initial ITC studies in Kümmel et al. are likely more physiologically relevant than the shorter CPX26-48 version.

*3) The criticism in Trimbuch of the original FRET results by Kümmel et al. is that the decreased FRET in the cases interpreted by Krishnakumar as indicating the complexin accessory helix angling away from the SNARE bundle could also be interpreted as increased flexibility in the structured state due to the mutations and truncations. The response of the current paper is to include new data with a FRET pair having a longer Rzero, showing more substantial elimination of FRET due to the GPGP insertion, for which the small but possibly detectable FRET change in the previous experiments with the shorter Rzero dye pair. Despite the fact that the new longer-Rzero, GPGP-insert data support Kümmel's interpretation that complexin angles away modestly, not totally flexible, the reviewers don't view these FRET experiment as conclusive evidence of either static or dynamic configurations. They don't have the distance or time resolution needed. Rather, they are useful observations that can be consistent with models derived from aggregating several other experiments*.

*In this context, the authors may wish to consult the paper by Choi et al, Structure, 2011 (10.1016/j.str.2011.01.011) that showed that complexin is in an extended conformation when bound to complete SNARE complex, whereas it is in a more coiled conformation when unbound. Thus, if the unbound part of complexin were completely unstructured when bound to the truncated SNARE complex, one would expect actually higher FRET efficiency to the label site on the SNARE complex. In contrast, a lower FRET efficiency is observed, consistent with the notion that complexin is in an angled conformation when bound to truncated SNARE complex. However, the mutation shown in*
Figure 4
*induces a further decrease of FRET efficiency, a result that is incompatible if complexin were unstructured (in the subsection headed “FRET Analysis”). Rather, this insertion probably introduces a hinge between the core and accessory domains, which may explain the decreased FRET efficiency. A discussion of these points should be included*.

The reviewers expressed reservation regarding our earlier conclusion that the new FRET experiment with probes with larger R0 value provides conclusive evidence that CPX_acc_ angles away from the SNARE complex and not totally flexible. We have included the calculated FRET distances (Figure 4) between CPX_acc_ and pre-fusion/post-fusion SNARE complex (with and without the GPGP insert) to further support our conclusion that CPX_acc_ angles away from the SNARE complex as a rigid helix and not completely flexible, and we have moderated our conclusions. In addition, as recommended by the reviewers, we incorporate the [4] paper, which suggests that a completely unstructured CPX would result in a higher FRET signal than a more structured form, to support that the accessory helix is in fact structured when CPX is bound to SNAREΔ60. With respect to the GPGP-insertion mutant, we have revised the text to read that the GPGP-insertion between CPX_cen_ and CPX_acc_ results “enhanced flexibility” of the CPXacc rather than complete un-structuring of the accessory helix (under the subsection headed “FRET Analysis”).

[Editors' note: further revisions were requested prior to acceptance, as described below.]

*1) The authors now show new and highly relevant DLS data, indicating that indeed translational diffusion decreases upon increasing the concentration of CPX-SNARE_delta60 (please clarify whether this DLS data is indeed for CPX or CPX26-83, used for panel A in this figure). In particular if the DLS data was obtained for CPX (Kd ∼15 microM), the DLS data are not particularly compelling as evidence for hetero-oligomerization involving multiple SNARE complexes and complexins, as one would have expected a substantially larger increase in apparent particle radius than observed. Thus, the authors are requested to present a model that might explain their observation*.

The reviewers requested to present a model to explain the DLS data showing the increase in particle size as a function of concentration. We have therefore included the following section to the Materials and methods section and the predicted particle size curve in Figure 5:

“In the DLS, there is oligomerization of CPX-SNAREΔ60. We can try to model this process as an equilibrium between bound and unbound CPX-SNAREΔ60, wherein the bound state is generated by the interaction of CPX_acc_ with the t-SNARE groove. […] This is reasonable considering the CPX-SNAREΔ60 as a prolate ellipsoid of dimensions 12 x 5 x 5 nm^3^, then the theoretical r_0_ would be (6 x 2.5 x 2.5)^1/3^ = 3.35 nm. Hence, the oligomerization we observe by DLS is perfectly consistent with the Kd values, we measure by ITC (Figure 2)”.

*2) If such a model can be obtained, an important control for the DLS experiment is missing: measuring the DLS signal for a tuncated CPX without the accessory helix in complex with SNARE_delta60. Ideally, the authors should also perform the experiment for the* “*poor-clamp*” *mutant (A30E, A31E, L41E, A44E)*.

The reviewers also requested to include control experiments in the DLS to verify that the CPX_acc_-t-SNARE interaction gives rise to the high order oligomers. We therefore included the following control experiments: (i) non-clamp mutant CPX26-83 (ncCPX26-83) carrying A30E, A31E, L41E, A44E mutations with SNAREΔ60 and (ii) wild type CPX26-83 in complex with fully assembled ternary SNARE complex. In both cases, we observe a small change in the average particle size, which could be fitted using our model, only assuming a very low affinity interaction (K_d_ = 250 µM for ncCPX into SNAREΔ60 and K_d_ = 300 µM for CPX into full ternary SNARE complex), such as arise from non-specific aggregation. The data and fitting are included in Figure 5 and referenced in the main text (under the subsection headed “NMR Analysis”).

*3) The very low ellipticity of ∼-7.3*10^-3 at ∼200 nm seen in the CD spectrum of CPX26-48 (*Figure 5*) is a concern. For example, synaptobrevin produces a minimum of ∼ -13 *10^-3 at ∼200 nm (J.Biol.Chem, 272, 28036-28041, 1997). The concentration measurement of CPX26-48 and normalization of the CD should be checked*.

The reviewers expressed concern regarding the low ellipticity values in the CD spectrum of CPX26-48. We believe that this due to difficulty in accurately measuring the concentration of a small peptide, which lacks any aromatic residues. However, the data is still consistent with the idea that the peptide corresponding to the accessory helix alone (CPX26-83) is not completely structured.

*4) Currently, the substantial number of Kd values and their corresponding proteins are distributed throughout the text and figures. It would be useful if they could also be compiled in a single table, such that a reader can more easily compare the various values, along with an explanation of the various constructs and SNARE complexes used, including* “*blocked*” *SNARE complex*.

As requested by the reviewers, we have now included a table (Table 1) summarizing all the K_d_ values and the corresponding protein constructs, including description of these complexes.

*5) In the subsection headed “NMR Analysis”, CD only defines total helicity of CPX26-83, and one cannot uniquely assign this to 26-48. Moreover, the NMR study of Pabst et al. (JBC 275:19808-18, 2000) provides fairly strong evidence that residues 28-64 in free complexin form one contiguous helix. Thus, the statement should be corrected accordingly*.

We agree with the reviewers that CD spectrum of CPX26-83 applies to the entire helix and cannot be uniquely applied to the accessory helix alone, and we have therefore modified the relevant section in the first paragraph under the subsection headed “NMR Analysis” to read “our circular dichroism (CD) measurements showed that CPXacc (residues 26-48) by itself is unfolded, but CPX26-83 forms a stable α-helical structure (Figure 5)”.